# Impact of Cognitive Impairment on Quality of Life in Multiple Sclerosis Patients—A Comprehensive Review

**DOI:** 10.3390/jcm13113321

**Published:** 2024-06-04

**Authors:** Sara Gómez-Melero, Javier Caballero-Villarraso, Begoña Mª Escribano, Alejandro Galvao-Carmona, Isaac Túnez, Eduardo Agüera-Morales

**Affiliations:** 1Maimónides Biomedical Research Institute of Córdoba (IMIBIC), 14004 Cordoba, Spain; sara.gomez.melero@gmail.com (S.G.-M.); am1esdub@uco.es (B.M.E.); agalvao@uloyola.es (A.G.-C.); itunez@uco.es (I.T.); doctoredu@gmail.com (E.A.-M.); 2Clinical Analyses Service, Reina Sofía University Hospital, 14004 Cordoba, Spain; 3Department of Biochemistry and Molecular Biology, Universidad of Córdoba, 14071 Cordoba, Spain; 4Department of Cell Biology, Physiology and Immunology, Faculty of Veterinary Medicine, University of Cordoba, 14071 Cordoba, Spain; 5Department of Psychology, Universidad Loyola Andalucía, 41704 Sevilla, Spain; 6Neurology Service, Reina Sofia University Hospital, 14004 Cordoba, Spain

**Keywords:** multiple sclerosis, disability, quality of life, cognitive impairment, neurodegenerative diseases

## Abstract

Multiple sclerosis (MS) is characterized by a variety of symptoms that have a major impact on quality of life (QoL) even in early stages. In addition to individual motor, sensory, visual disturbances, and brainstem and sphincter disorders, which are expressed through the widely used Expanded Disability Status Scale (EDSS), other manifestations of MS have a detrimental effect on overall functioning and quality of life, such as cognitive impairment, depression, anxiety, fatigue, and pain. However, when talking about QoL, categorical definitions cannot be used because although the concept is generally understood, it is highly nuanced. Suffering from MS can significantly reduce QoL. Numerous research studies have focused on trying to identify and assess which are the elements that most affect the loss of QoL in MS people. However, in addition to the fact that the measurement of QoL can be subjective, it is very difficult to consider these elements in isolation, as they are interrelated. One such limiting factor of QoL that has been investigated is cognitive impairment (CI). This has been shown to have an impact on the lives of MS people, although the different approaches that have been taken to assess CI have evident limitations.

## 1. Introduction

Multiple sclerosis (MS) is an autoimmune, inflammatory disease of the central nervous system (CNS) characterized by inflammation, demyelination, and axonal degeneration [1]. Its prevalence has increased worldwide since 2013 and it is estimated that in 2020 there were 2.8 million people with the disease [2]. It is the leading cause of non-traumatic disability in young adults. In women it usually has an earlier onset than in men, and women are two to three times more at risk of developing it than men [3,4]. Generally, the first symptoms of MS can be detected between the ages of 20 and 40 years, and appear in childhood in less than 1% of sufferers, although the prevalence of late-onset MS (debut above the age of 50 years) ranges from 4 to 9.4 [5,6]. MS is more common in colder regions and among Caucasians [7]. Although the etiology of the disease is uncertain, studies suggest that environmental factors, such as Epstein–Barr virus infection, and genetic factors, such as those related to immune function and activation, are involved in MS [1,8].

MS is a heterogeneous disorder and can manifest differently in each individual. It can show a wide range of clinical symptoms including spasticity; motor impairment; cognitive impairment; visual and sensory loss; bladder, bowel, and sexual dysfunction; fatigue; and mood disorders [9]. This symptomatology may affect their daily activities, family, and social life, and may compromise their long-term independence [10,11].

Disease progression varies greatly from individual to individual, whereby MS is classified into four main types: relapsing remitting multiple sclerosis (RRMS), primary progressive multiple sclerosis (PPMS), secondary progressive multiple sclerosis (SPMS), and progressive relapsing multiple sclerosis (PRMS). In most cases, the disease initially follows a relapsing–remitting course in which it shows clear exacerbations of neurological symptoms, known as flares, followed by a remission phase without disease progression. The RRMS form accounts for 85% of MS forms and is usually followed in frequency by SPMS, characterized by a deterioration of symptoms, with a reduction in relapses and no apparent signs of remission. It is estimated that 30-50% of patients suffering from RRMS eventually develop its secondary progressive form (SPMS). In PPMS (10% of the total), symptoms progressively worsen without relapses or clearly defined periods of remission [12]. There are 3–5% of patients with progressive–recurrent MS (PRMS), in which there is a gradual deterioration with overlapping relapses [5]. Finally, in 2012, clinically isolated syndrome (CIS) was added as a new disease course, referring to a first episode of inflammatory demyelination in the CNS that could develop into multiple sclerosis if additional activity occurs [13]. In other words, it could be considered as a pre-MS form that may eventually develop into MS [14].

The diagnosis of MS is based on the presence of clinical manifestations, together with an MRI (magnetic resonance imaging) study in which cerebral demyelinating plaques are detected. For confirmation, visual, auditory brainstem, and somatosensory evoked potentials can be added, as well as cerebrospinal fluid (CSF) analysis, in which characteristic oligoclonal bands can be observed in the electrophoretic study, which are due to intrathecal synthesis of antibodies [12]. Within the currently available therapeutic arsenal are the so-called disease-modifying therapies (DMTs), which can be immunomodulatory or immunosuppressive. These are mainly used to treat RRMS in order to prevent disease progression but are not as effective in the case of progressive clinical forms (PPMS and SPMS), and treatment remains limited in these patients [15,16].

Globally, the average age of diagnosis is 32 years and there is no curative treatment, so it is a chronic disease that can affect a person’s life in many ways, especially during their peak years of productivity. For this reason, one of the aims of treatment is to optimize as far as possible the quality of life (QoL) of patients and to maintain functionality in aspects important to them [17]. The gradual increase in physical and cognitive disability among people with MS has substantial effects that negatively impact their social, economic, and individual well-being [1], and the aim of this review is to compile the available scientific literature on the impact of MS-specific cognitive impairment on QoL. Most studies on this topic include many factors that may influence QoL, but the problem with addressing these factors in a global way is that it is not possible to know the impact of each factor in isolation; it also carries the risk that some of the factors included in these studies may actually be potential confounding factors. There is a paucity of scientific literature focusing, in particular, on the impact of cognition on QoL in MS. Therefore, among the various elements related to this disease, we pay special attention to the specific influence of cognitive function on QoL in patients with MS.

## 2. Course of Multiple Sclerosis

### 2.1. Relapsing Remitting Multiple Sclerosis (RRMS)

In the majority of cases, the disease starts as RRMS. In this phase, inflammation in the CNS, caused by both adaptive and innate immune system components, results in myelin destruction and gradual neurological decline, usually progressing in successive flares that manifest as a brief episode of acute neurological dysfunction, followed by complete or partial remission and the subsequent resolution of symptoms [18]. Activated immune cells cause CNS lesions leading to symptoms such as visual disturbances, tingling and numbness, fatigue, uro-genital disorders, spasticity, and memory disorders [5]. As an inflammatory disease, the onset of symptoms during an RRMS flare-up usually develops gradually over several days. A typical initial presentation of RRMS is a unilateral optic neuritis characterized by a gradual onset of monocular visual loss, pain in eye movement, and impaired color vision [19].

### 2.2. Secondary Progressive Multiple Sclerosis (SPMS)

Almost 65% of RRMS patients subsequently develop SPMS, which is considered the second stage of RRMS. Thus, these patients experience SPMS 10–15 years after the onset of RRMS, in which inability advances independently of relapses. In this situation, acute exacerbations become less common, but the patient experiences a gradual decline in neurological dysfunction, with or without intervals of remission, suffering increasing disability, which may (but not always) be accompanied by new brain lesions [18]. Many patients often have increased weakness, urinary system disorders, fatigue, stiffness, mental disorders, and psychological impairment [5].

### 2.3. Primary Progressive Multiple Sclerosis (PPMS)

Around 10–15% of MS patients receive a diagnosis of PPMS, which largely affects the spinal cord tracts. Patients with PPMS usually have fewer brain lesions. Derived symptoms include walking problems, weakness, stiffness, and balance disorders [5]. In PPMS, symptoms are expected to appear gradually and insidiously for at least 12 months after diagnosis [19]. PPMS is the most disabling clinical form [15].

### 2.4. Progressive Relapsing Multiple Sclerosis (PRMS)

PRMS is the least common type of MS, occurring in approximately 5% of patients. This condition is characterized by a gradual worsening of the condition from the onset, similar to PPMS. There are occasional relapse episodes of intensified symptoms similar to those experienced in RRMS [5].

### 2.5. Clinically Isolated Syndrome (CIS)

Although recognized for some time, it was not until 2012 that CIS was established as an evolving variant that can sometimes develop into MS. The term CIS refers to a first clinical event that is highly suggestive of CNS demyelinating disease, but does not yet meet with sustained activity over time to be diagnosed as clinically definite MS. The symptoms present usually evolve acutely or sub-acutely over days or weeks, and involve the optic nerve, spinal cord, brainstem, or cerebellum. As with other MS attacks, the episode is expected to last at least 24 h and to occur in the absence of fever or infection [13].

## 3. Medical History of MS

MS is characterized by demyelinating lesions in regions such as the periventricular white matter, subcortical area, infratentorial location, or even in the spinal cord. Histopathology also reveals widespread involvement of the cerebral grey matter, although this is not well appreciated on conventional MRI [19]. The clinical manifestations depend on the areas of the brain or spinal cord affected, and therefore, depending on the location of the lesions, patients may develop any neurological sign or symptom, including the aforementioned motor, sensory, and cognitive impairments [20].

Symptoms are generally unpredictable and uncertain, varying greatly from patient to patient and within patients over time. During the course of MS, some abnormalities appear to be more dominant or have a greater effect on functional ability. Eventually, symptoms such as social, occupational, and psychological complications may appear [5].

The World Health Organization developed an International Classification of Functioning, Disability, and Health (ICF) [21] that defines a common language to describe the impacts of the disease at different levels, which are impairment (problems in anatomical or physiological structures and their symptoms and signs), activity limitation (formerly known as “disability”), participation restriction (formerly known as handicap), and contextual factors (environmental and personal). People with MS may have a combination of deficits, such as physical, cognitive, psychosocial, behavioral, and environmental problems. Classified according to the WHO ICF, these involve impairments (strength, coordination, balance, spasticity, memory, urinary urgency), resulting in function limitation (mobility, self-care, incontinence, pain, cognitive deficits) and restricted social participation (impact on work, driving, family, finances). A study utilizing the brief ICF core set for MS, which encompasses 20 categories related to bodily functions, activities and participation, and environmental factors, revealed that the ICF category “moving (d445)” was the most diminished activity, observed in 75% of participants (with 51% of participants experiencing severe impairments). However, there is a lack of studies assessing the completeness of the brief core set of ICF for MS in clinical practice [22].

According to the evidence, cognitive impairment (CI) is a condition that occurs in all MS phenotypes, and the prevalence differs according to the criteria and study population. The relationship between CI and MS in the early stages is not well established. However, it is claimed that the beginning of the disease predicts earlier conversion from RRMS to SPMS and, subsequently, disability progression. Therefore, an accurate diagnosis of CI in the early stages of the disease can be used as a determinant for disease progression. Furthermore, there are findings that reveal that patients with SPMS appear to experience higher CI compared to patients with CIS, RRMS, and PPMS [23].

## 4. Assessment and Disease Scoring Systems

As suggested above, the diagnosis of MS is based on clinical manifestations and the presence of demyelinating lesions on neuroimaging tests. Among these clinical manifestations, the characteristics of the flares should be observed, in addition to the history and physical examination findings. The key diagnostic principle is dissemination in time (DIT) and dissemination in space (DIS). There is no specific laboratory test to diagnose MS [19]. Grading is conducted according to the disability of MS, as it plays a key role in determining effective therapeutic approaches and, consequently, scoring scales are used for this purpose [24].

### 4.1. Expanded Disability Status Scale (EDSS)

The Expanded Disability Status Scale (EDSS), published in 1983, is currently the most widely used scale and was one of the first standardized tools for the assessment of MS-related disability [25]. The EDSS is a comprehensive scale that assesses various functional systems, encompassing pyramidal functions (muscle strength, tone, and reflexes), cerebellar functions (coordination and balance), brainstem functions (eye movements, speech, and swallowing), sensory functions (fine touch, pain, and vibratory sensitivity), bowel and bladder functions, visual function, brain functions (cognition), and ambulation [24,26]. It consists of an ordinal scale ranging from 0 (normal neurological examination) to 10 (death due to MS) and does not require much more than a complete neurological examination [26]. At the lower end of the scale (0–3.0), the score is based on abnormalities detected during the neurological examination, and at the upper end of the scale (7.5–9.5), it depends on basic functions and the capacity to perform activities of daily living [27]. Although the EDSS is a commonly accepted clinical tool for assessing disease progression, it has a low sensitivity on the higher values of the scale that focus on general QoL and self-care. Therefore, there are considerable functional measures, such as cognitive abilities, visual function, sensorimotor function, and bowel and bladder function, that are not as well detected [9].

### 4.2. EDSS-Plus

Despite physical disability, between 40% and 65% of patients with progressive MS experience CI, impacting their QoL more significantly than physical impairment [28]. However, this is often difficult to measure using EDSS alone [27]. In these patients, the EDSS appears to be less sensitive in detecting clinically significant factors contributing to disability progression, especially sensory-motor dysfunctions of the upper extremities, as well as lower extremities (with the Timed 25-Foot Walk Test (T25FW) assessing gait) and cognitive dysfunctions. Instead of the EDSS scale, it has been recommended to use the EDSS-Plus scale to assess the progression of patients with SPMS and PPMS [29]. The EDSS-Plus tool consists of a combination of the EDSS score, the T25FW, and the Nine-Hole Peg Test (9HPT) [29]. The T25FW assesses the time required for the patient to walk the 25-Foot Walk (7.62 m) distance in a habitual manner safely, quickly, and without pausing. The 9HPT assesses manual dexterity by rating the patient’s ability to insert and remove nine pins from holes in a box [18,26]. The EDSS-Plus score is twice as sensitive as the EDSS alone in assessing disability progression in patients with SPMS (59.5% vs. 24.7%) [29].

### 4.3. Multiple Sclerosis Functional Composite (MSFC)

The Multiple Sclerosis Functional Composite (MSFC) is another scoring measure that combines the T25FW as a measure of ambulation, the 9HPT as a measure of upper extremity function, and the Paced Auditory Serial Addition Test (PASAT) as a measure of cognitive function [27]. The PASAT evaluates auditory information processing ability with the presentation of 60 numbers every 3 s to the patient. The task involves adding each new digit to the one immediately preceding it [26]. Thus, the test assesses auditory information, processing speed, and calculation skills, although it also does not appear to be very sensitive when assessing impaired cognitive function in SPMS [30].

### 4.4. Neuropsychological Batteries

For cognitive assessment, several tools are available, and the use of neuropsychological batteries is recommended following a positive screening test, subjective cognitive complaints reported by the patient or caregiver, or incongruity between clinical perception and screening evaluations, or in specific socio-occupational situations. The most commonly used neuropsychological batteries in MS are the Neuropsychological Screening Battery for Multiple Sclerosis (NSBMS), the Brief International Cognitive Assessment for MS (BICAMS), and the Minimal Assessment of Cognitive Functioning in Multiple Sclerosis (MACFIMS) [31].

The NSBMS was one of the first neuropsychological batteries to assess MS-related CI, introduced in 1990 by neuroscientists from the Cognitive Function Study Group in the USA. This battery is composed of the Selective Reminding Test (SRT) to assess verbal learning and memory, the 7/24 Spatial Recall Test (SPART) to assess visual memory acquisition and delayed recall, the PASAT to assess attention maintenance and information processing speed, and the Word List Generation Test (WLGT) to assess spontaneous word utterance and mental agility. Shortly afterwards, the same group developed another simplified battery called the Brief Repeatable Battery of Neuropsychological Tests (BRB-N). This was accompanied by the SDMT and uses the 10/36 SPART instead of the 7/24 version [32,33,34]. Through this screening tool, a characteristic CI pattern was defined in MS affecting memory. Information processing efficiency, executive functions, attention, and processing speed are other most commonly compromised functions [34].

The MACFIMS is considered an enhanced version of the BRB-N, as it assesses all cognitive functions impacted by MS. MACFIMS incorporates tasks assessing visuospatial function, such as the Benton Line Orientation Judgment (JOLB), the PASAT at a rate of 2 s, and the Delis-Kaplan Executive Function System (D-KEFS) test for executive function [31].

Currently, the BICAMS is the most popular international battery used to assess CI in patients with MS, measuring mental processing speed and memory. The international expert consensus committee has recommended this test as a short study battery for cognitive assessment in MS. It takes only 15 min to complete the BICAMS assessment and is therefore potentially feasible in clinical practice [33,35].

In addition, the TRAIL Making Test is a component of many neuropsychological test batteries and provides information on visual search, scanning abilities, processing speed, executive functions, and mental agility. The TRAIL test consists of two parts, Test A (TRAIL-A) and Test B (TRAIL-B), the latter requiring the subject to connect circled numbers and letters in ascending and alternating order. The score represents the amount of time required to complete the task [36]. The TRAIL-B test is generally regarded as a test of executive function, specific to rapid shifting of cognitive sets and divided attention [37].

### 4.5. Neuorimaging

It has been demonstrated that patterns of brain activation are different in MS samples compared to healthy controls during cognitive task performance [38,39]. Although researchers still consider neuropsychological assessments the benchmark for assessing changes in cognitive function, many studies are now incorporating additional outcome evaluations, specifically imaging parameters. One of the most frequently used paraclinical tools for diagnosing and predicting the course of MS is MRI, which enables in vivo visualization of MS-related pathology and anatomical damage [40].

The impact of disruptions in normal-appearing white matter (WM) on cognitive function has been validated through various imaging techniques, from magnetization transfer and diffusion-based methods to quantitative MRI. Investigating abnormalities in normal-appearing grey matter (GM) in vivo has been enabled by advanced MRI techniques, including magnetization transfer imaging, quantitative relaxometry, quantitative susceptibility mapping, and diffusion MRI [41].

It has been shown that MS patients with cognitive decline exhibit more pronounced GM atrophy in the right anterior cingulate cortex and bilateral supplementary motor area. Additionally, they have reduced resting state functional connectivity in the right hippocampus of the right working memory network and in the right insula of the default mode network [41,42]. Several diffusion MRI studies suggest a decrease in topological efficiency (a measure of network integration) within the structural connectome of MS patients. Additionally, a more random organization of the GM network is associated with poorer cognitive functioning, regardless of GM atrophy [41]. Cortical and deep GM atrophy appears to be crucial in isolated CI. Early identification and customized patient interventions, particularly for IPS, might assist in delaying additional cognitive deterioration [43].

### 4.6. Other Tests to Assess the Neurocognitive Impact of MS

An additional diagnostic instrument for evaluating of cognitive functions in MS patients, which also assesses concentration capacity, attention maintenance, and oculomotor speed, is the Symbol Digit Modalities Test (SDMT) [18]. The SDMT is a simple substitution task in which the patient matches numbers with symbols, where the answers can be given orally or in writing. The SDMT takes less than 5 min and can be performed at the patient’s own pace, so it is less stressful for patients than the PASAT. Similarly, the Low-Contrast Letter Acuity (LCLA) test can support the assessment of visual function, a potential weakness of the original MSFC. A recent study corroborated that the addition of the SDMT and LCLA to the MSFC improves its psychometric characteristics as a multidimensional composite [44]. The SDMT exhibits greater sensitivity compared to the PASAT, and is better at distinguishing RRMS from SPMS than alternative neuropsychological evaluations [45,46]. However, recent research proposes that SDMT scores improve with follow-up, likely due to repetition by patients, and so do not accurately reflect the gradual cognitive decline seen in SPMS patients [47]. Consequently, arguably, the best diagnostic tool for monitoring disease progression in patients with SPMS would be one that includes the use of several tests that assess deterioration of clinical status at various levels. In any case, the EDSS currently remains the most common scale in routine clinical practice and is frequently the only one used [18].

## 5. QoL in MS

### 5.1. QoL Measures

Clinicians are generally more accustomed to clinical outcome measures based on findings obtained by healthcare professionals, but patient-based measures, called Patient Reported Outcomes (PROs), can be successfully used to characterize the disease from different perspectives taking into account the patient’s point of view [26]. The inclusion of PROs for the follow-up of MS is gaining acceptance, as they allow patients to share crucial information with clinicians on how they experience the impact of the disease, the effects of treatment, and the issues that are of most concern to them. There are many PRO tools available designed to measure factors affecting patients with MS, including symptoms, limitations of daily activities, fatigue, global QoL, and health-related quality of life (HRQoL) [9].

The World Health Organization (WHO) defined quality of life in 1995 as “a person’s perception of his or her position in life in the context of the culture and value systems in which he or she lives and in relation to his or her goals, expectations, norms and concerns. It is a broad concept that is affected in complex ways by a person’s physical health, psychological state, level of independence, social relationships and relationship to salient features of his or her environment” [48].

HRQoL is a discrete component of QoL; however, a universally accepted definition has not yet been established. It is a concept that is used to represent a person’s perception of their health and is measured through standardized tools and instruments, providing a quantitative method for monitoring individual health status in response to intervention changes over time [49]. Measurement of the different components of HRQoL is crucial, particularly during clinical evaluation and therapy planning. It is important to utilize cross-culturally valid measures to perceive variations between socio-cultural groups in order to ensure appropriate treatment guidelines [50].

Over the past two decades, HRQoL assessments have been progressively incorporated into research studies of neurodegenerative disorders, including MS [51]. MS affects the HRQoL of patients at all stages of the disease. There are several measures to assess QoL in patients with MS, some of which are generic or specific to MS, while others combine both [49].

The SF-36 questionnaire is a commonly utilized tool for assessing self-perceived HRQoL. This measure is not specific to any disease, age, or treatment group, and is a short and complete questionnaire [27]. The objective of the SF-36 is to capture the perspectives of the person on his QoL, thus allowing the lived experience to be present. It is a test that works well at the group level, consisting of eight subscales that measure different aspects of health: (1) physical functioning; (2) role restrictions due to physical health troubles; (3) body pain; (4) general health perceptions; (5) social activity; (6) role limitations due to emotional problems; (7) vitality (energy/fatigue); and (8) overall mental health (psychological distress and well-being) [52]. From the eight subscales, two synthetic components can be derived: the physical component summary (PCS) and the mental component summary (MCS), where the PCS comes from the subscale 1–4 and the MCS comes from the subscale 5–8. PCS and MCS were built to simplify and enhance the analysis of results [27]. The SF-36 questionnaire has been verified in the population with MS, and it has been shown that the components of the physical summary present a high correlation with EDSS [53]. A cross-sectional study by Nortvedt and collaborators concluded that the SF-36 survey could capture all effects of MS on a person’s QoL [54].

The creation of the Multiple Sclerosis Quality of Life-54 (MSQOL-54) item inventory aimed to fulfil the necessity for HRQoL measures to be used in research of quality of care and clinical effectiveness in MS. Compared to other tools, its main advantage is that it integrates a generic approach and a disease-focused approach. In fact, MSQOL-54 is a multidimensional MS-specific HRQoL assessment tool, based on generic SF-36 supplemented with 18 MS-specific items [51]. MSQOL-54 comprises 54 questions, 18 of which are in 14 domains specific to patients with MS and 36 are related to general QoL [55].

MSIS-29 is a clinically useful and scientifically sound measurement of the physical and psychological impact of MS, suitable for clinical trials and epidemiological studies. MSIS-29 exhibits a strong correlation with other well-known self-reported measures, such as SF-36. The items are scored on a Likert scale from 1 to 5. MSIS-29 gives a total score ranging between 29 and 145, with higher scores indicating a greater impact of the disease or lower QoL [56,57].

Another MS-specific QoL measure that has good internal consistency and test–retest reliability is the Multiple Sclerosis International Quality of Life questionnaire (MusiQoL) [58]. The MusiQoL questionnaire is a self-administered and multidimensional survey. It was developed from patient interviews and was specifically designed to reflect patients’ perspectives on how MS affects their daily lives. The different areas related to MS that are evaluated are daily life activities, psychological well-being, symptoms, personal relationships, sentimental and sexual life, adaptability, and rejection [59].

### 5.2. MS Factors That Affect QoL

As a chronic disease, MS significantly affects all facets of patients’ lives. Therefore, evaluating QoL and the repercussions of functional and mental status on it is as essential for clinicians as it is for patients themselves. This evaluation helps to take measures to improve and stabilize the physical and mental state of patients, allowing them to improve the way they perceive their lives [60]. However, evaluating and analyzing QoL in patients with MS (PwMS) is a demanding task since it is a complex and, by nature, subjective concept [61].

Several authors have identified, over the years, various factors that negatively influence HRQoL in these patients such as low household income, a high EDSS score, a low score in the 9HPT test, a diminished verbal responsiveness, and a debilitating and progressive disease course. All of these factors have been linked to lower HRQoL, and it has been shown that lower HRQoL portends worse survival in PwMS [61,62,63].

Symptoms of MS are highly variable, and because of them about half of PwMS require ongoing care and support. However, advances in treatment in recent years have allowed many to live independently and control their symptoms [64]. In contrast, some patients may continue to experience decreased QoL despite successful disease management [65]. Although most factors contribute directly to the economic burden of the disease (for example, through treatment or costs of sick leave), it remains uncertain whether they equally affect patients’ subjective reports on QoL and whether this differs between the remitting–recurrent and progressive MS phenotypes [60]. There is a need for greater recognition of the presence and effects of these factors, as well as effective treatment options for specific deficiencies, which could improve QoL and reduce indirect costs arising from unemployment, healthcare and home care teams, home modifications related to disability, and personal care [65].

It has been found that certain factors that contribute negatively to QoL in MS can be depression, fatigue, mood disorders, cognition, work situation, and personality and behavioral changes [66]. In the study conducted by Yalachkov and collaborators, higher levels of QoL in MS were associated with university studies, shorter duration of the disease, reduced levels of depression and psychological distress, and a course of disease with relapses and remissions compared to a progressive course [60]. In addition, during relapse periods, there is a lower QoL compared to during remission periods [67].

In addition to clinical health status, a number of complex social, psychological, and contextual factors influence patients’ QoL, and identifying these factors is an important objective when considering how best to support PwMS [68]. Studies show that people with MS experience a reduction in QoL due to psychosocial challenges such as sexual dysfunction, problems with defecation, feelings of impotence and shame, and social stigma [55]. In addition, physical disability and loss of upper limb function can significantly limit daily activities and have been shown to correlate with lower QoL measurements [63].

Similar findings have been observed for fatigue and concomitant psychopathological disorders, such as depression, which exert a negative influence on the QoL of patients [60]. Patients with depression have significantly lower QoL scores with respect to health perception, sexual dysfunction, mental health, general QoL, emotional dysfunction, and limitations due to emotional problems [69]. Fatigue is a disabling symptom of MS that hinders patients’ physical functions and employability. The two most-used questionnaires to evaluate it are the Fatigue Severity Scale (FSS), which focuses mainly on physical aspects, and the Modified Fatigue Impact Scale (MFIS), which covers physical, psychosocial, and cognitive aspects [70]. Fatigue can contribute to various disabilities and restrictions of activity and social activities as a result of MS affecting QoL [71].

Predictably, the variety and severity of symptoms and associated impairment seem to play a crucial role in QoL. Fatigue, CI, and pain are the subjects of a large number of studies and are confirmed as important risk factors. Longitudinal studies suggest that increased fatigue, pain, and CI also lead to worse QoL. This has important clinical implications, as priority should be given to the treatment of these symptoms [72].

Both CI and neuropsychological reserves (i.e., factors such as a high educational level and better premorbid intelligence, which, according to the hypothesis, preserve cognition despite brain damage) are also related to QoL [73]. According to different studies, the prevalence of IC is 43 to 70% among patients with MS and it is known that it occurs more frequently in progressive forms of the disease. This deterioration is a frequent but underdiagnosed symptom in MS. Men are more susceptible to cognitive decline than women and, although higher education and lower EDSS scores appear to be protective factors in women, they are not really protective. This is possibly due to the faster erosion of cognitive reserves due to a higher rate of brain atrophy in men [72]. The routine assessment of cognitive status and the detection of its deterioration are of great importance because they can help decide on the best therapy for PwMS [74]. Several studies recognize a variety of cognitive affectations as risk factors affecting PwMS QoL, such as cognitive fatigue, memory loss, and planning/organization dysfunction. It has been shown that maintaining executive functioning is a protective factor for QoL [72].

## 6. Relationship of QoL and CI in PwMS

CI is highly prevalent in MS and generally reduces HRQoL [75]. It is well documented that CI has a negative and dramatic impact on various aspects of QoL, regardless of the degree of physical disability, affecting the activities of daily social and professional life [76,77]. The cognitive aspects most frequently affected are the information processing speed (IPS), complex attention, working memory, visuospatial capacity, and executive functions, with predominance of disorders of executive functions in progressive forms of MS and an amnesic profile in the sender-recurrent MS [31].

Like all symptoms of MS, CI varies greatly in severity and progression. While some patients may not experience any decline or do so gradually, others may undergo significant deterioration. Some changes can be relatively mild and easily compensated for, while others can affect performance in key facets of daily life, such as work or driving. Cognitive deficits have a potential impact on daily functioning and QoL, demonstrating the need to detect deficits, assess their severity, and remedy cognitive challenges that impair functioning [78].

A study conducted to determine relevant predictors of QoL from an extensive list of symptoms showed that cognitive aspects accounted for approximately 21% of the total variance in MSIS-29 QoL scores. The results of the linear regression carried out in this study indicate that motor skills and memory are significant predictors of QoL [57]. They also linked a higher educational level and higher performance in cognitive tasks with a lower impact of the disease and a higher QoL, as demonstrated in previous research [60,79,80]. The need to assess and diagnose CI, and subsequently treat and/or help people, compensates for these impairments in the workplace is fundamental to improving QoL.

Social cognition encompasses the cognitive mechanisms underlying interpersonal abilities like social perception, empathy, and “theory of mind”, which refers to identifying the mental states of others through interpreting their facial expression, other behavioral signs, and social context. Recognized as a crucial aspect of daily QoL and involved in various psychological processes, social cognition is one of the deficits observed as part of the CI process in MS that affects the QoL and social activity of patients. Findings from one study demonstrate a link between social cognition deficits and diminished social and psychological quality of life, regardless of the duration or severity of the disease, age, or the results of formal neurocognitive tests [81].

There are several studies that positively correlate HRQoL with high scores in cognitive aspects in PwMS. Strober and collaborators found that SDMT has a significant association with measures that evaluate HRQoL and psychological functioning. In addition, they found a significantly greater association between the SDMT and the PCS of the SF-36 than between the PASAT and the PCS [46].

Another study found that MSFC scores correlate with SF-36 scores and provide QoL information regardless of EDSS scores. This may be due to the inclusion of the PASAT CI test in the MSFC. The EDSS has been criticized for its lack of precision, using an ordinal scoring system, being insensitive to changes in time, and failure to capture significant aspects of the disease such as cognitive functions [82].

The study carried out by Goldman and collaborators calculated the correlation between functional measures (T25FW, 9HPT, SDMT, and LCLA) and QoL measures self-reported by the participants. At the beginning of the study, the T25FW, 9HPT, and SDMT tests were moderately correlated with the PCS component of the SF-36 and significantly, albeit weakly, with the MCS component. The LCLA test was weakly correlated with both components of SF-36. Among participants with worsening from start to finish in T25FW, 9HPT, or SDMT, the average PCS component of SF-36 also worsened [44].

Another study also explored the correlation between individual measures of physical impairment and cognitive function and the PCS and MCS components of SF-36. This study showed that MCS correlates with PASAT, TRAIL-B, and SDMT cognitive function measures. In addition, they obtained a significant relationship between the 9HPT for the non-dominant hand and the TRAIL-B and SDMT measurements. All cognitive tests used in this study, conducted by Højsgaard Chow et al., were correlated with SF-36 MCS, while there was no correlation between cognitive tests and PCS [27].

In patients with early MS, the available knowledge on the association between cognitive scores and HRQoL measures is limited. Glanz et al. studied this association by conducting a multivariate study in which they checked for depression, since they saw a strong correlation between depression and HRQoL in the patients in the study. Scores on IPS tests showed significant connections with various measures of HRQoL, including physical well-being, fatigue, and social support. Significant correlations were observed between the PASAT and the PCS of the SF-36. IPS has been identified as a key deficit in MS, and its association with HRQoL may have important implications for rehabilitation. These observations were noted in patients with limited CI and minimal physical disability. These results suggest that cognitive rehabilitation programs aimed at improving cognitive abilities can also improve the QoL of PwMS in their early stages [83].

Jakimovski et al. investigated how depression scores influence the connection between deep GM volume and physical limitations and life satisfaction in PwMS. They found that lower scores in the Emotional Well-Being and Thinking/Fatigue subscales of HRQoL were associated with a higher lesion load. Similarly, the MSQOL-54 and Short Form-36 questionnaires showed that measures of pain, cognition, and overall QoL were linked to larger lesion volumes and reduced global brain volumes [40].

Improving cognitive skills is directly related to improving daily functions [84]. However, this link is rarely discussed. Cognitive rehabilitation interventions have been criticized because changes in cognitive functions similar to those of trained subjects can be observed without changes in daily functions [38,85]. In fact, most of the studies conducted in PwMS that examined the effectiveness of a rehabilitative approach did not examine whether improvement in underlying cognitive processes is generalized to everyday life [28,38].

In a recent Cochrane review on memory rehabilitation in MS, evidence was reported to support the effectiveness of memory rehabilitation not only on memory function, but also on QoL. However, the authors criticized the findings of the reviewed studies and called for robust randomized clinical trials with rigorous methodological standards and improved quality of reporting, using ecologically valid outcome assessments to measure treatment generalization [38]. Ecological validity is the degree to which clinical tests of cognitive functioning predict functional decline and is an area of interest in neuropsychology [86]. Ecological assessment has emerged because classical assessment of neuropsychological tests is performed under artificial and simple conditions with few or no distractors. Because the environment of the classic evaluation of neuropsychological tests is controlled, real conditions are not reflected, so ecological tools seem interesting for the detection of subtle cognitive deficits and their consequences in conditions of daily life. These tools could be attractive to patients who can easily understand problems and anticipate the future. Neuropsychological testing and ecological testing are complementary tools to assess cognition in PwMS [87]. Rouaud et al. demonstrated the ability of ecological tests to detect executive dysfunctions in PwMS in real conditions of daily life that were undervalued by classical neuropsychological test batteries [88]. Similarly, it has been observed that in PwMS there is an impairment of “dual tasks”, presenting difficulties in carrying out two or more activities simultaneously [89,90].

The study conducted by Biernacki and collaborators evaluated the possible predictors of HRQoL in patients with MS. They found that CI in men is a negative factor in general QoL, in the limitation of roles due to emotional issues, in sexual functions, and in the domain of satisfaction with sexual functions. The reason why CI only influenced HRQoL in men but not in women can be due to the higher prevalence of CI found in men than in women [61].

A person with CI is not able to accurately identify the present problem, and therefore usually has a better score on the overall HRQoL assessment, but to some extent feels that something has changed, and the CI may have a negative impact on the overall QoL. A sudden drop in social activity, either real or only perceived, due to a debilitating disease, can greatly diminish the HRQoL of a PwMS [61].

A scientific consensus has not yet been reached on the contribution of cognitive and physical disability to reduced HRQoL. One study indicates the importance of disease characteristics not related to physical aspects, by showing that the EDSS does not correlate with the MSQOL-54 [63]. It has been suggested that CI in patients with MS is a prognostic indicator for the development of the disease and, consequently, for greater disability and more deficits in daily and social functioning. A statistically significant association between specific aspects of CI and QoL has been observed from a health perspective, both mental and physical, and therefore more attention should be paid to CI in PwMS [23]. A better understanding of the cognitive deficits of MS will inform the emerging field of cognitive rehabilitation, which aims to restore cognitive functioning (often through intensive cognitive training programs), or teach compensatory strategies to attenuate the deleterious effects of the deficits and refractory cognitive effects on QoL [35].

The purpose of this comprehensive review is to provide an update on the interrelationships between the brain involution of MS, its neurocognitive repercussions, and how these influence the quality of life of these patients. In summary, from the above we can deduce the following: (i) MS is related to multiple manifestations that impair the quality of life of patients; (ii) these manifestations are closely interrelated and may influence each other; and (iii) the ways of assessing CI may have severe limitations.

We believe that future research should focus on devising methods to try to measure CI with more specific and precise methodologies, which would allow CI to be assessed in isolation and, therefore, provide a better understanding of the impact of CI on QoL of MS patients. This knowledge may be a good starting point to propose strategies to improve QoL of MS patients, as well as to evaluate the gain in QoL of patients in whom the progression of CI is controlled.

## 7. Limitations

Several studies have been conducted on the impact of CI on QoL in PwMS and the results revealed are contradictory. While many studies show that CI is a major determinant of HRQoL in patients, other studies based on self-reported HRQoL measures surprisingly do not find that CI is a significant factor, so the current results on this subject are controversial [61,66,82,91].

The discrepant findings reported in the literature may be the result of heterogeneous PwMS samples in terms of disease course (recurrent vs. progressive), disease duration, and disability, as well as differences in cognitive and HRQoL measures [83]. The diversity of degrees of disability suffered by PwMS can significantly affect the evaluation of their QoL, both in physical and mental health [10]. There is frequently a mismatch between the clinician’s and patient’s perceived importance of different facets of HRQoL. Some doctors prioritize physical aspects over mental and emotional aspects in comparisons between patients [58]. Most tools for assessing QoL rely on self-report, so the results, despite being quantified measures, often remain a matter of interpretation, and it is therefore difficult to objectively associate CI and QoL in PwMS. Furthermore, significant variability can be observed in the results of the literature when psychopathological symptoms are examined in relation to the different tools used (with different duration, depth, or main focus) [61]. Although the merit of the studies carried out is unquestionable, the independent contribution of various cognitive dysfunctions and clinical disabilities in relation to QoL is rarely investigated, limiting their possible association [61].

Another possible reason for the discrepancy between the investigations may be attributed to the unrealistic self-reporting of patients with severe CI who are often unaware of their deficit, and sometimes even affected by euphoria [61]. CI appears to be primarily predictive of what PwMS are actually capable of, rather than what they can report [66]. Distinguishing cognitive decline caused by MS from the cognitive effects of treatable comorbidities (such as depression) is challenging. Clinical practitioners should therefore conduct assessments with formal measuring tools; the National Multiple Sclerosis Society (NMSS) recommends cognitive testing every 1 or 2 years. Since patients may not be aware of their own CI, it is also recommended that family members make their own observations [65].

In addition, the weaker relationship of CI with HRQoL could be due to the cognitive reserve capacity of patients. Patients who have greater active and passive cognitive reserve capacity tend to report lower levels of perceived disability, are healthier, and experience higher levels of well-being in PRO surveys [73].

The underlying reason for the inconsistency of these results is multifactorial and there are several reasons that limit the study of the association between QoL and CI. Among these factors we should consider the existence of two situations that are present in many PwMS, namely fatigue and depression. Both situations can considerably limit Qo. Therefore, to determine the impact of CI in isolation on QoL in these patients, cases of MS that occur with CI should be separated from those who also have fatigue and/or depression, or studies should proceed to a stratified analysis of patients (in the presence or absence of these situations) [92,93,94].

## 8. Conclusions

MS causes considerable disruption to the lifestyle, social situation, employment status, and annual income of affected individuals, negatively impacting their personal, occupational, and social functioning, and thereby reducing their overall QoL. Although physical disability is important for the performance of daily activities, it cannot explain the extent of difficulties that PwMS encounter in many daily activities, especially in tasks requiring significant cognitive effort. Evaluation and treatment of PwMS should be performed on a much broader spectrum than physical and MRI examinations, addressing a preventative approach. Currently efforts to improve cognitive status in MS are in the early stages and future work is crucial for improving the well-being of PwMS. Pharmacological and behavioral studies to improve cognitive function in MS are scarce and future research is needed in which optimal research strategies are applied to allow recommendations for pharmacological therapy or behavioral interventions for cognitive deficits in MS. From this review, it can be concluded that CI is a clinical condition that negatively affects QoL of PwMS and, although great progress has been made in identifying effective treatments for cognitive deficits, there is still much to do to optimize the cognitive rehabilitation potential of PwMS and its improvement in QoL.

## Data Availability

No new data were created or analyzed in this study. Data sharing is not applicable to this article.

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
