# Peer review of "Impact of Cognitive Impairment on Quality of Life in Multiple Sclerosis Patients—A Comprehensive Review"

_jcm, 2024, doi:10.3390/jcm13113321_

Round 1
Reviewer 1 Report
Comments and Suggestions for Authors
Comment 1: the type of the review (comprehensive) should be mentioned on the title. If it not a comprehensive review, the structure of the article should be changed and other sections added
Comment 2: i think that a piece of discussion should be added before the limitations and that the researchers should mention where their research ends up, what its usefulness is, what future research would help in this area and what they suggest
Author Response
1) The type of the review (comprehensive) should be mentioned on the title. If it not a comprehensive review, the structure of the article should be changed, and other sections added.
Done. Many thanks to the reviewer for noticing and bringing it to our attention.
2) I think that a piece of discussion should be added before the limitations and that the researchers should mention where their research ends up, what its usefulness is, what future research would help in this area and what they suggest.
Done. We have highlighted it in red.
“The purpose of this comprehensive review is to provide an update on the interrelationships between the brain involution of MS, its neurocognitive repercussions and how these influence the quality of life of these patients. In summary, from the above we can deduce that: i) MS is related to multiple manifestations that impair the quality of life of patients. ii) These manifestations are closely interrelated and may influence each other. iii) The ways of assessing CI may have severe limitations.
We believe that future research should focus on devising methods to try to measure CI with more specific and precise methodologies, which would allow this CI to be assessed in isolation and therefore provide a better understanding of the impact of CI on the QoL of MS patients. This knowledge may be a good starting point to propose strategies to improve the QoL of MS patients, as well as to evaluate the gain in QoL of patients in whom the progression of CI is controlled”.

Reviewer 2 Report
Comments and Suggestions for Authors
The manuscript is well organized and covers the QoL MS literature in an easily digestable way.
My sole comment is whether the authors/editors have considered a need to evaluate the relevant neuroimaging literature on QoL or on QoL-linked cognitive impairments in MS. There is a lot of literature linking both clinical and cognitive impairments and cognitive fatigue in MS to specific neuroimaging markers (e.g. functional connectivity differences in MS patients correlating with cognitive/clinical impairments and fatigue).
If this has been considered and left out as a conscious decision for the scope of the paper, this comment can be disregarded. However, I feel that it warrants at least a mention as it shows a physiological cause of these impairments and may then suggest a link between MS, impairments and QoL through an avenue not completely evaluated by the authors.
Author Response
The manuscript is well organized and covers the QoL MS literature in an easily digestible way.
My sole comment is whether the authors/editors have considered a need to evaluate the relevant neuroimaging literature on QoL or on QoL-linked cognitive impairments in MS. There is a lot of literature linking both clinical and cognitive impairments and cognitive fatigue in MS to specific neuroimaging markers (e.g. functional connectivity differences in MS patients correlating with cognitive/clinical impairments and fatigue).
If this has been considered and left out as a conscious decision for the scope of the paper, this comment can be disregarded. However, I feel that it warrants at least a mention as it shows a physiological cause of these impairments and may then suggest a link between MS, impairments and QoL through an avenue not completely evaluated by the authors.
Authors’ replay:
We greatly appreciate this observation and agree with this reviewer, because this neuroimaging perspective can offer relevant information to the potential reader of the manuscript. Incorporating this idea, we have added five references. Therefore, it has been clarified in the revised manuscript. The information added in the manuscript has been marked with red color. In page 6 the following information has been added:
“4.5. Neuroimaging.
It have been demonstrated that patterns of brain activation are different in MS samples compared to healthy controls during cognitive task performance [38], [39]. Although researchers still consider neuropsychological assessments the benchmark for assessing changes in cognitive function, many studies are now incorporating additional outcome evaluations, specifically imaging parameters. One of the most frequently used paraclinical tools for diagnosing and predicting the course of MS is MRI, which enables in vivo visualization of MS-related pathology and anatomical damage [40].
The impact of disruptions in normal appearing white matter (WM) on cognitive function has been validated through various imaging techniques, from magnetization transfer and diffusion-based methods to quantitative MRI. Investigating abnormalities in normal appearing grey matter (GM) in vivo has been enabled by advanced MRI techniques, including magnetization transfer imaging, quantitative relaxometry, quantitative susceptibility mapping, and diffusion MRI [41].
It has been shown that MS patients with cognitive decline exhibit more pronounced GM atrophy in the right anterior cingulate cortex and bilateral supplementary motor area. Additionally, they have reduced resting state functional connectivity in the right hippocampus of the right working memory network and in the right insula of the default mode network [41], [42]. Several diffusion MRI studies suggest a decrease in topological efficiency (a measure of network integration) within the structural connectome of MS patients. Additionally, a more random organization of the GM network is associated with poorer cognitive functioning, regardless of GM atrophy [41]. Cortical and deep GM atrophy appears to be crucial in isolated CI. Early identification and customized patient interventions, particularly for IPS, might assist in delaying additional cognitive deterioration [43].”
In page 11 the following information has been added:
“Jakimovski et al. investigated how depression scores influence the connection between deep GM volume and physical limitations and life satisfaction in pwMS. They found that lower scores in the Emotional Well-Being and Thinking/Fatigue subscales of HRQoL were associated with a higher lesion load. Similarly, the MSQOL-54 and Short Form-36 questionnaires showed that measures of pain, cognition, and overall QoL were linked to larger lesion volumes and reduced global brain volumes [40].”

Round 2
Reviewer 1 Report
Comments and Suggestions for Authors
All revisions were addressed